# Standard Behaviour of $Bi_2Sr_2CaCu_2O_{8+\delta}$ Overdoped

**Giovanni Alberto Ummarino** [1,2] 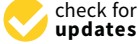

1   Istituto di Ingegneria e Fisica dei Materiali, Dipartimento di Scienza Applicata e Tecnologia, Politecnico di Torino, Corso Duca degli Abruzzi 24, 10129 Torino, Italy; giovanni.ummarino@polito.it
2   Moscow Engineering Physics Institute, National Research Nuclear University MEPhI, Kashirskoe hwy 31, 115409 Moscow, Russia

**Abstract:** I calculated the critical temperature and superconducting gap in the framework of one band d wave Eliashberg theory with only one free parameter in order to reproduce the experimental data relative to $Bi_2Sr_2CaCu_2O_{8+\delta}$ (*BSCCO*) in the overdoped regime. The theoretical calculations are in excellent agreement with the experimental data and indicate that cuprates in the overdoped regime are well described by standard d-wave Eliashberg theory with coupling provided by antiferromagnetic spin fluctuations.

**Keywords:** high Tc superconductors; Eliashberg equations



## 1. Introduction

The properties of cuprates, and of $Bi_2Sr_2CaCu_2O_{8+\delta}$ (*BSCCO*) in particular, depend strongly on their oxygen content [1,2]. This tunability allows one to study the doping dependence of superconductivity—an approach that has been used to investigate the fundamental properties of several unconventional materials [3–5]. For more than 30 years, there has been a major debate in the scientific community about which mechanism is responsible for superconductivity in cuprates. However, most of the research has been focused on the underdoped regime, where there is a variety of competing mechanisms that probably do not specifically concern the superconducting state but that contribute to confuse and hide the true mechanism of superconductivity. Several studies have underlined the differences between these new, high critical temperature superconductors and the old, low critical temperature superconductors whose behavior is perfectly explained by BCS theory or its natural generalization: Eliashberg theory. In this paper, however, I aim to highlight the commonalities of cuprates with the old superconductors, regarding not the coupling mechanism but the overall theoretical framework. In a recent paper [6], the authors experimentally investigated the behavior of *BSCCO* in the overdoped regime by measuring the critical temperature ($T_c$), the superconducting gap value ($\Delta_0$), the electron boson coupling constant ($\lambda_Z$) and the representative energy ($\Omega_0$) of the mechanism responsible for superconductivity. Such an experimental study is very useful for theoreticians trying to clarify the mechanism responsible for superconductivity in these materials. Moreover, in the overdoped regime, there are superconductors with very high critical temperatures (in our case, $T_c \leq 91$ K) which make the study interesting in itself. In [6], the authors show angle-resolved photoemission spectroscopy studies of *BSCCO* in the overdoped region up to the to non-superconducting phase. They find that the coupling strength, $\lambda_Z$, in the antinodal region of the Fermi surface weakens with doping and at the critical value $\lambda_{Zc} \simeq 1.3$, the superconductivity disappears. This is the evidence that in the overdoped regime, superconductivity is determined primarily by the coupling strength, which is probably connected with antiferromagnetic spin-fluctuations. I have tried to reproduce these very interesting experimental data in a comprehensive model .

In Figure 1, the experimental data present in the paper are shown relative to different values of doping $p$ (the doping away from half-filling) where the doping is expressed as

$p = 2A_{FS} - 1$ and $A_{FS}$ is the area enclosed by the Fermi contour [6], which are used as input parameters in the Eliashberg equations. It is possible to see that both quantities of the electron boson coupling constant ($\lambda_Z$) and the representative energy ($\Omega_0$) decrease with the increase of doping. For comparison with $\lambda_Z$, the calculated coupling $\lambda_\Delta$ in the gap channel, as explained after, is also shown in the figure.

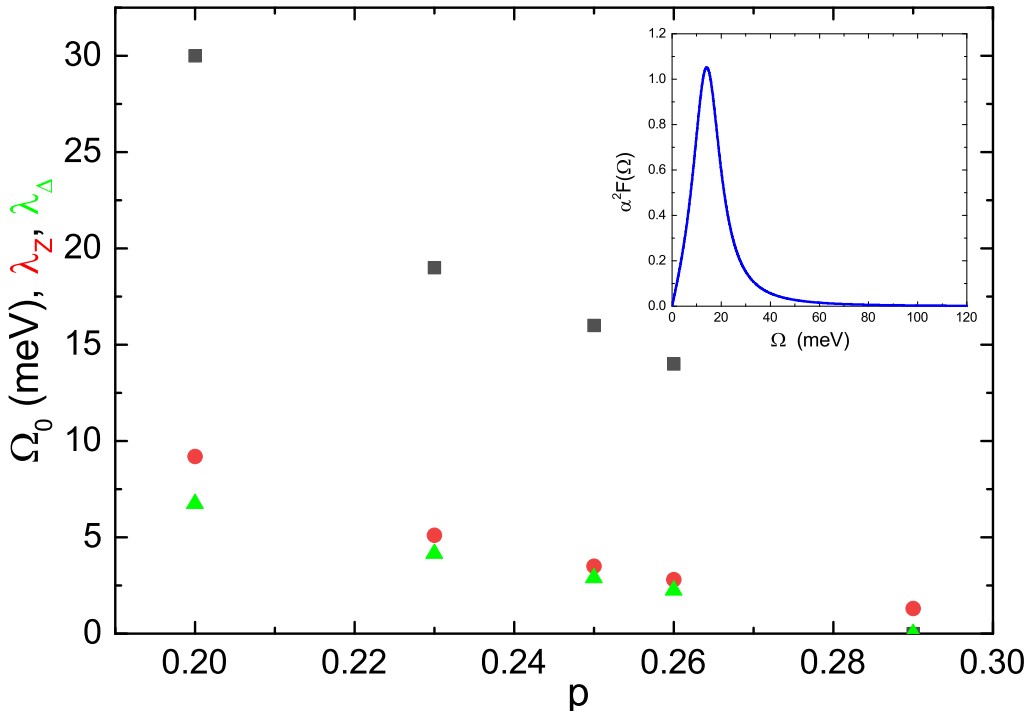

**Figure 1.** Experimental input parameters of Eliashberg theory: representative energy $\Omega_0$ (black full squares) and electron-boson coupling constant $\lambda_Z$ (red full circles) and $\lambda_\Delta$ (green full triangles) as a function of doping $p$. $\lambda_\Delta$ is the value obtained, via Eliashberg equations, to reproduce exactly the experimental critical temperatures. In the inset, the electron–boson spectral function for the last value of doping $p = 0.26$ is shown.

## 2. Model

I calculated the experimental critical temperatures and the superconductive gaps shown in Figure 2 by solving the one band d-wave Eliashberg equations [7–14]. In this case, two coupled equations for the gap $\Delta(i\omega_n)$ and renormalisation functions $Z(i\omega_n)$ have to be solved ($\omega_n$ denotes the Matsubara frequencies). The d-wave one-band Eliashberg equations in the imaginary axis representation are

$$\omega_n Z(\omega_n, \phi) = \omega_n + \pi T \sum_m \int_0^{2\pi} \frac{d\phi'}{2\pi} \Lambda(\omega_n, \omega_m, \phi, \phi') N_Z(\omega_m, \phi') \tag{1}$$

$$Z(\omega_n, \phi) \Delta(\omega_n, \phi) = \pi T \sum_m \int_0^{2\pi} \frac{d\phi'}{2\pi} \left[\Lambda(\omega_n, \omega_m, \phi, \phi') - \mu^*(\phi, \phi')\right] \times$$
$$\times \Theta(\omega_c - |\omega_m|) N_\Delta(\omega_m, \phi') \tag{2}$$

where $\Theta(\omega_c - \omega_m)$ is the Heaviside function, $\omega_c$ is a cut-off energy and

$$\Lambda(\omega_n, \omega_m, \phi, \phi') = 2 \int_0^{+\infty} \Omega d\Omega \alpha^2 F(\Omega, \phi, \phi') / \left[(\omega_n - \omega_m)^2 + \Omega^2\right] \tag{3}$$

$$N_Z(\omega_m, \phi) = \frac{\omega_m}{\sqrt{\omega_m^2 + \Delta(\omega_m, \phi)^2}} \qquad (4)$$

$$N_\Delta(\omega_m, \phi) = \frac{\Delta(\omega_m, \phi)}{\sqrt{\omega_m^2 + \Delta(\omega_m, \phi)^2}} \qquad (5)$$

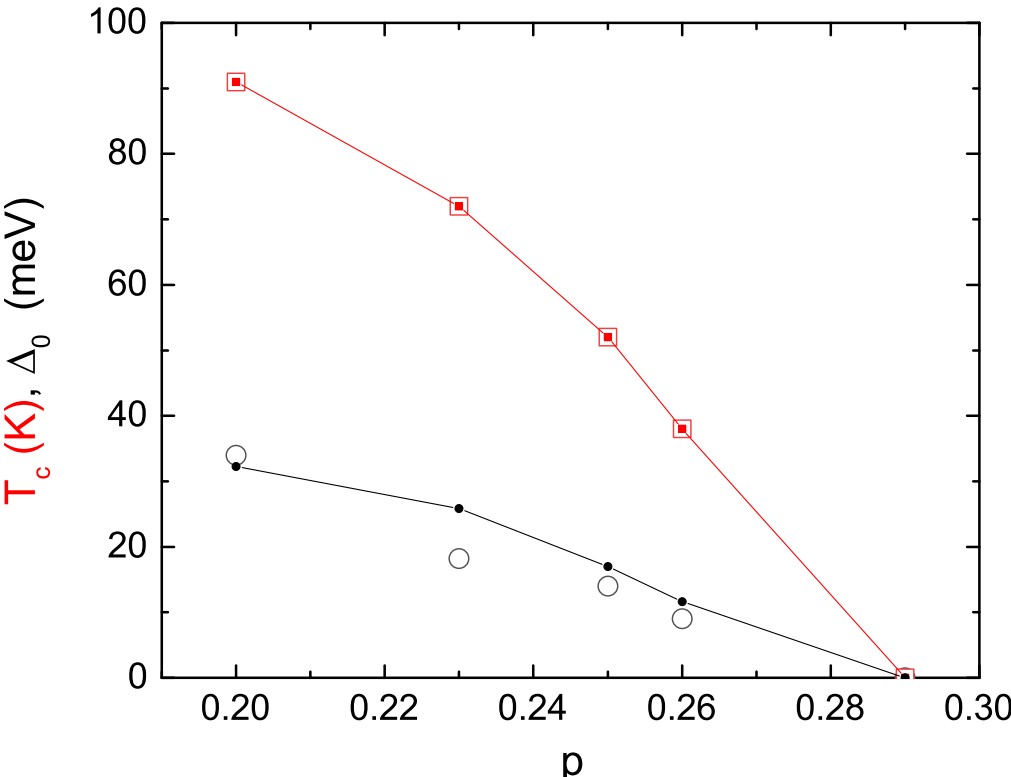

**Figure 2.** Calculated values of critical temperature (full red squares) and superconductive gap (full black circles) compared with experimental data (open red squares for the critical temperature and open black circles for the superconductive gap) in the function of the doping *p*. The lines are visual guides.

I assume [7–14] that the electron–boson spectral function $\alpha^2(\Omega)F(\Omega, \phi, \phi')$ and the Coulomb pseudopotential $\mu^*(\phi, \phi')$ are at the lowest order to contain separated s and d-wave contributions,

$$\alpha^2 F(\Omega, \phi, \phi') = \lambda_s \alpha^2 F_s(\Omega) + \lambda_d \alpha^2 F_d(\Omega)\sqrt{2}cos(2\phi)\sqrt{2}cos(2\phi') \qquad (6)$$

$$\mu^*(\phi, \phi') = \mu_s^* + \mu_d^*\sqrt{2}cos(2\phi)\sqrt{2}cos(2\phi') \qquad (7)$$

as well as the self energy functions:

$$Z(\omega_n, \phi) = Z_s(\omega_n) + Z_d(\omega_n)cos(2\phi) \qquad (8)$$

$$\Delta(\omega_n, \phi) = \Delta_s(\omega_n) + \Delta_d(\omega_n)cos(2\phi) \qquad (9)$$

The spectral functions $\alpha^2 F_{s,d}(\Omega)$ are normalised in such a way that $2\int_0^{+\infty} \frac{\alpha^2 F_{s,d}(\Omega)}{\Omega}d\Omega = 1$; of course, in this model, $\lambda_Z = \lambda_s$ and $\lambda_\Delta = \lambda_d$ because in this case I search for solutions of the Eliashberg equations with a pure d-wave form, as indicated by the experimental data, for the gap function $\Delta(\omega, \phi') = \Delta_d(\omega)cos(2\phi)$ (the s component is zero, and this happens for example when [15] $\mu_s^* >> \mu_d^*$). In the more general case, $\lambda_\Delta$ has d and s components. The renormalization function $Z(\omega, \phi') = Z_s(\omega)$ has only

the $s$ component because the equation for $Z_d(\omega)$ is a homogeneous integral equation whose only solution in the weak-coupling regime is $Z_d(\omega) = 0$ [16]. For simplicity, we also assume that $\alpha^2 F_s(\Omega) = \alpha^2 F_d(\Omega)$ and that the spectral functions are the difference of two Lorentzians; i.e., $\alpha^2 F_{s,d}(\Omega) = C[L(\Omega + \Omega_0, Y) - L(\Omega - \Omega_0, Y)]$ where $L(\Omega \pm \Omega_0, Y)) = [(\Omega \pm \Omega_0)^2 + (Y)^2]^{-1}$, $C$ is the normalization constant necessary to obtain the proper values of $\lambda_s$, $\Omega_0$ and Y are the peak energies and half-width, respectively. The half-width is $Y = \Omega_0/2$. This choice of the shape of spectral function is a good approximation of the true spectral function [17] connected with antiferromagnetic spin fluctuations. The same thing also happens in the case of iron pnictides [18]. In any case, even making different choices for Y, it can be verified that as Y increases, the value of $\lambda_d$ decreases and there are no large $\lambda_d$ variations for reasonable Y choices. For example, if $Y = \Omega_0$, the reduction of $\lambda_d$ is of the order of four percent with respect to $Y = \Omega_0/2$. The trend as a function of $\lambda_d$ doping remains the same; only the coefficients in the function fit ($\lambda_d$ versus $\lambda_s$) change slightly. The cut-off energy is $\omega_c = 1000$ meV and the maximum quasiparticle energy is $\omega_{max} = 1100$ meV. In the first approximation, we put $\mu_d^* = 0$ (if the $s$ component of the gap is zero, the value of $\mu_s^*$ is irrelevant).

In this model, the experimental input parameters are $\Omega_0$ and $\lambda_s$ while there is only one free parameter: $\lambda_d$. I solve the imaginary axis d-wave Eliashberg equations for each value of $\Omega_0$ and $\lambda_s$ and I seek the value of $\lambda_d$ to obtain the correct critical temperature that represents the most reliable experimental data, which can be more precisely measured than the value of superconductive gap. After, via Padè approximants [19], I calculate the low-temperature value ($T = 4$ K) of the gap because, in the presence of a strong coupling interaction, the value of $\Delta_d(i\omega_{n=0})$ obtained by solving the imaginary-axis Eliashberg equations can be very different from the value of $\Delta_d$ obtained from the real-axis Eliashberg equations [10,20]. This approach to reproduce experimental data has proved to be very efficient and successful with several materials [21,22].

## 3. Results and Discussion

In Figure 2, the results are shown. From Figure 2, it is possible see that the critical temperatures are perfectly reproduced and the behaviour of gap with the doping is reproduced well enough, which is remarkable in the framework of a very simple model without any "exotic" factors. However, the large values of the coupling constant for doping values close to optimal doping may be perplexing. Probably the values of the coupling constants are effective values [23] because, in this model, I do not take into account the violation of the Migdal theorem [24,25] which almost certainly happens. In fact, it has been shown that using an Eliashberg theory generalized to the case in which the Migdal theorem is not valid, one obtains values of the coupling constants $\lambda_\Delta$ and $\lambda_Z$ that are much smaller than those used in the standard Eliashberg theory to produce the same critical temperatures [26]. I find that the link between $\lambda_d$ (free parameter, determined by the calculus of critical temperature exactly equal to experimental one) and $\lambda_s$ (experimental data, input parameter) (see Figure 1) is reproduced very well by the equation

$$\lambda_d = 1.738(\lambda_s - 1.325)^{0.657} \tag{10}$$

Furthermore, from this formula, it is possible to see that the critical value of $\lambda_s$ is $\lambda_{sc} = 1.325$ from the experimental data [6]. The calculation of the critical temperature from the solution of Eliashbeg equations in this case depends strongly on the values of $\lambda_d$ and $\lambda_s$, and also small differences in the values of coupling constants produce large variation in the critical temperature. For this reason, in the formula of $\lambda_d$, there are three decimal digits. In Figure 3, the values of $\Delta_d(i\omega_{n=0})$ as a function of temperature are shown.

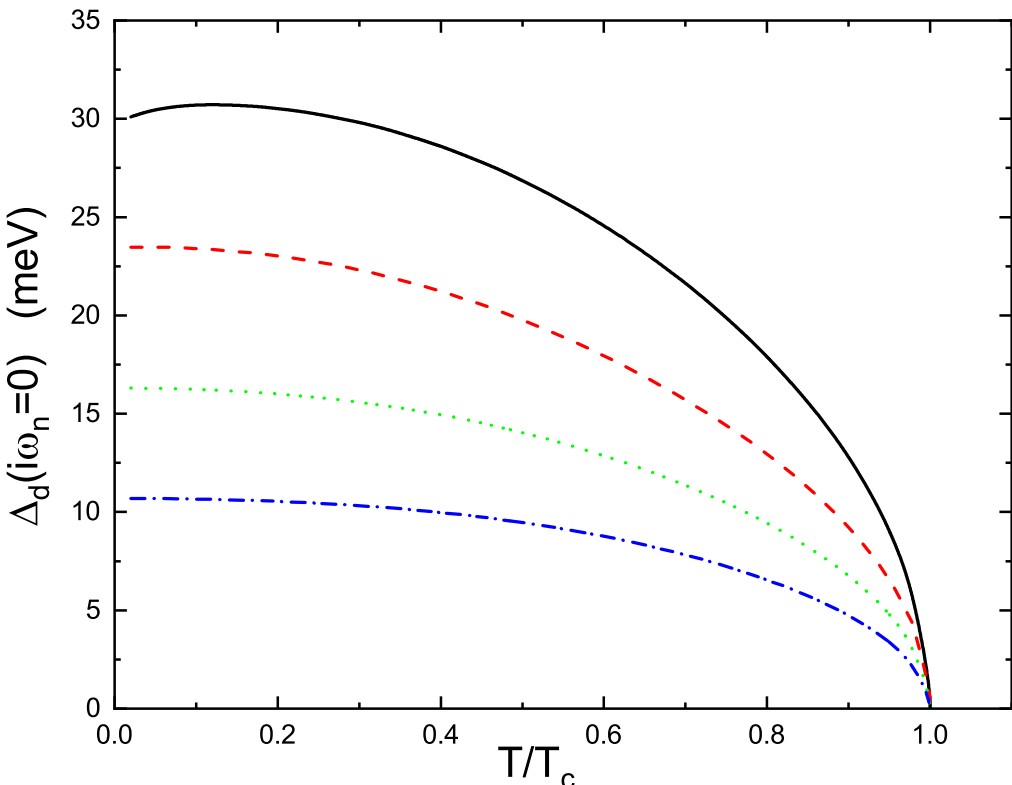

**Figure 3.** (Color online) Calculated values of $\Delta_d(i\omega_{n=0})$ as a function of normalized temperature $(T/T_c)$ for the four cases examined: $p = 0.20$, black solid line; $p = 0.23$, red dashed line; $p = 0.25$, green dotted line; and $p = 0.26$, dark blue dashed-dotted line.

### 4. Conclusions

In this paper, I have shown that the experimental data ($T_c$ and $\Delta_0$) in the overdoped regime for *BSCCO* can be reproduced by a very simple model: the standard band d-wave Eliashberg equations with coupling provided by antiferromagnetic spin fluctuations. This indicates that the superconducting state has no particular characteristics in the overdoped regime. The fact that this simple model explains the experimental data very well could be a new stimulus for further theoretical investigations in the underdoped regime that neglects a large number of exotic competing orders present in the normal state, which probably are not directly connected with the superconducting phase.

**Funding:** This research received no external funding.

**Institutional Review Board Statement:** Not applicable.

**Informed Consent Statement:** Not applicable.

**Data Availability Statement:** Not applicable.

**Acknowledgments:** The author acknowledges support from the MEPhI Academic Excellence Project (Contract No. 02.a03.21.0005) and D. Torsello for useful suggestions.

**Conflicts of Interest:** The authors declare no conflict of interest.

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
