# Peer review of "Standard Behaviour of Bi2Sr2CaCu2O8+δ Overdoped"

_condensedmatter, doi:10.3390/condmat6020013_

Round 1

Reviewer 1 Report

In this paper the Author analyzes the experimental data of Ref. [9]
for overdoped BSCCO within the context of the anisotropic Eliashberg theory.
Taking a selected set of the experimental data as input, namely the
boson characteristic energy, the superconducting critical temperature Tc
and the electron-boson dimensionless coupling
governing the band renormalization in the normal state,
and with a reasonable choice of the electron-boson coupling magnitude
in the superconducting channel dictated by the TC itself,
he shows that also other independent superconducting relevant
quantities, i.e. the low-temperature superconducting gap,
can be estimate in a striking agreement with the experimental
findings.
This provides a strong evidence of the validity of the Eliashberg
framework in the overdoped phase of BSCCO.
The paper is sound and convincing, so I recommend publication.

I suggest however the Author to correct some typos and mispelling
and few inaccuracies that might affect the readability:

  • line 14: "ha" -> "has";
    - line 15: "there are a variety" -> "there is a variety";
    - lines 41-42: in describing the contents of Fig. 1, it is mentioned
       that there are shown: Tc, Delta_0, lambfa_Z and Omega_0.
       However, different quantities are shown there: Omega_0, lambda_s, lambda_d.
       The text and the contents of the figure shoudl match;
    - line 85: "raison" -> "reason"
    - in the caption of Fig. 2 it is stated that calculated values (filled
        symbols) of Tc and Delta_0 (squares and circles, respectively) are
        displayed. However the experimental data for Tc (open red squares)
        are not actually visible. This is not surprising since the
        theoretical model parameters are chosen to reproduce exactly the
        experimental Tc. However the lack of the mentioned open red
       squares is confusing. I suggest to rephrase, and maybe to simplify,
       the caption.
    - line 96: "fase" -> "phase"

Author Response

First of all I would like to thank the referee for their constructive advice and observations.

line 14: "ha" -> "has"; 
This error was corrected

- line 15: "there are a variety" -> "there is a variety";
This error was corrected

- lines 41-42: in describing the contents of Fig. 1, it is mentioned
   that there are shown: Tc, Delta_0, lambfa_Z and Omega_0.
   However, different quantities are shown there: Omega_0, lambda_s, lambda_d.
   The text and the contents of the figure shoudl match;
Yes, I agree with the referee, now the text and the contents of the figure matchs.

- line 85: "raison" -> "reason"
This error was corrected

- in the caption of Fig. 2 it is stated that calculated values (filled
    symbols) of Tc and Delta_0 (squares and circles, respectively) are
    displayed. However the experimental data for Tc (open red squares)
    are not actually visible. This is not surprising since the
    theoretical model parameters are chosen to reproduce exactly the
    experimental Tc. However the lack of the mentioned open red
   squares is confusing. I suggest to rephrase, and maybe to simplify,
   the caption.
Now in the figure the open red squares are larger than before and are visible.

- line 96: "fase" -> "phase"
This error was corrected

Reviewer 2 Report

This paper report on Migdal-Eliashberg (ME) calculations on the critical temperature, gap, and effective el-ph coupligs with some input parameters of overdoped BSSCO.

The analysis is sound and the author finds that in the overdoped regime, the experimental data BSSCO can be explained in terms of standard ME theory, with coupling constants that become smaller as the system moves towards higher doping values.

I have only a couple of minor remarks:

1) lines 38-44: the text referring to Fig. 1 does not match with the quantities displayed in the figure and reported in the caption.

2) In the "results and discussion" section, the author claims that the experimental data are reproduced by the simple model without "exotic" values of the parameter. Well, actually the values of the coupling constants at small doping (p=0.2-0.23) are a bit exotic as they are larger than 5. This is a very strong coupling regime where ME is probably not adequate. The author may expand on this point, clarifying why he thinks that these values have nothing of exotic (maybe adding some relevant reference?).

Note also that in the same paragraph (4th line) the author probably meant "effective values" instead of "efficient values".

Author Response

First of all I would like to thank the referee for their constructive advice and observations.

1) lines 38-44: the text referring to Fig. 1 does not match with the quantities displayed in the figure and reported in the caption.
The text was modified in this way: "It is possible to see that both quantities:
the electron boson coupling constant (LZ) and the representative energy (W0) decrease with the increasing of doping. For comparison with LZ, the calculated coupling LD in the gap channel, as it will be explained after, is also shown in the figure."

2) In the "results and discussion" section, the author claims that the experimental data are reproduced by the simple model without "exotic" values of the parameter. Well, actually the values of the coupling constants at small doping (p=0.2-0.23) are a bit exotic as they are larger than 5. 
This is a very strong coupling regime where ME is probably not adequate. The author may expand on this point, clarifying why he thinks that these values have nothing of exotic (maybe adding some relevant reference?).

Yes it is true. I add this explanation: " What may create some perplexity are the large values of coupling constant for doping values close to optimal doping [1]  
because in our model I do not take into account the violation of the Migdal theorem [2]  that almost certainly happens. In fact, it has been shown that using an Eliashberg theory generalized to the case in which the Migdal theorem is not valid, we obtain values of the coupling constants LD and LZ that are much smaller than those used in the standard Eliashberg theory to produce the same critical temperatures [3]. This mean that the theory reproduces well the experimental data but the values of coupling constant are effective and nor real. Real values are smaller"

[1] P.Benedetti, C. Grimaldi, L. Pietronero and G. Varelogiannis,  Europhys. Lett 28, 251 (1994).
[2]  L. Pietronero, S. Strassler, and C. Grimaldi, Phys. Rev. B 52, 516 (1995);
C. Grimaldi, L. Pietronero, and S. Strassler, Phys. Rev. B 52, 530 (1995).
[3]  G.A.Ummarino and R.S. Gonnelli, Phys. Rev. B 56, 14279 (1997).

The ref [1] is new. In ref [1] the concept of effective coupling is discussed while in ref [3] the effective coupling is quantified for some materials where the Migdal theorem doesn't work.

Note also that in the same paragraph (4th line) the author probably meant "effective values" instead of "efficient values".
Yes, this error was corrected.

Reviewer 3 Report

The article “Standard behaviour of Bi2Sr2CaCu2O8+δ overdoped” is worth publishing, but there are some sections that need clarification to make it more useful to others.  

I will admit from the start that I am an experimentalist, so there may be notations and conventions that I am not familiar with, but hopefully my comments will make the manuscript more accessible to a lager audience. 

My first difficulty is with Fig. 1.  Two different notations are used in the manuscript. In the Figure, λs and λd are written on the y-axis label, but in the main text these are referred to as λZ and λ.   In Section 2 (line 56) it does say “and of course in this model λZ = λs and λ = λd.” This statement needs to be earlier in the text to avoid confusion with Fig. 1.  I also wonder, it this just a change in notation or is there any physics involved? The manuscript implies these are parameters that come from ref. 9, although that reference seems to only measure one coupling parameter, and I was assuming that is λZ.   λ is apparently calculated, and needs to be defined or explained.  I also note that Tc is not included in Fig. 1 as implied in the text(line 41).  

After Equation 5 the phrase “I suppose” probably should be “I assume.”  Pleas check the meaning of that sentence.

After line 80, the word “the” before Fig. is not needed. 

In the abstract it reads that “cuprates in the overdoped regime are well described by standard d-wave Eliashberg theory with antiferromagnetic spin fluctuactions” (extra “c” in fluctuations). In the main text it is implied that Ω and Y characterize the spin fluctuations and these are fixed parameters in the fitting procedure. I can see that  Ω needs to be fixed because it was measured, but why does Y = 1/2, a simple even fraction, work fine for what would seem to be a sensitive parameter? 

Finally, the fit of the theory to the data is excellent, with only one free parameter.  Does this imply that λd is determined from the fit?  And yet it is related to λZ, which was measured. Something seems to be circular here, or in a way it shows the validity of Equation 10.  The conclusion is not clear to me. 

With these clarifications I believe I would be able to better understand the manuscript.  

Author Response

First of all I would like to thank the referee for their constructive advice and observations.

My first difficulty is with Fig. 1.  Two different notations are used in the manuscript. 
In the Figure, λs and λd are written on the y-axis label, 
but in the main text these are referred to as λZ and λ∆.   
Yes, this misprint was corrected in this way: "It is possible to see that both quantities: the electron boson coupling constant (LZ) and the representative energy (W0) decrease with the increasing of doping. For comparison with LZ, the calculated coupling LD in the gap channel, as it will be explained after, is also shown in Fig. 1."

The y-label in the figure was corrected.

In Section 2 (line 56) it does say “and of course in this model λZ = λs and λ∆ = λd.” 
This statement needs to be earlier in the text to avoid confusion with Fig. 1.  I also wonder, it this just a change in notation or is there any physics involved? 
Yes I use this notation because I solve the Eliashberg equations in the pure d-wave case while in general λ∆ has a d and s component but in the BSCCO overdoped the s component is absent so I can simplify the theory.

The manuscript implies these are parameters that come from ref. 9, 
although that reference seems to only measure one coupling parameter, and I was assuming that is λZ.
Yes, because λZ exists and is measured also for a value of doping where the material is not superconductive while in this case  λ∆ is zero because is the coupling connected just with the superconductive state.

λ∆ is apparently calculated, and needs to be defined or explained. 
λ∆ is a free parameter and is determined when I fix the calculate critical temperature equal to experimental one.

I also note that Tc is not included in Fig. 1 as implied in the text(line 41).  
Yes, the text was corrected

After Equation 5 the phrase “I suppose” probably should be “I assume.”  Pleas check the meaning of that sentence.
Yes, this error was corrected.

After line 80, the word “the” before Fig. is not needed. 
Yes, this error was corrected.

In the abstract it reads that “cuprates in the overdoped regime are well described by standard d-wave Eliashberg theory with antiferromagnetic spin fluctuactions” (extra “c” in fluctuations). 
Yes, this error was corrected.

In the main text it is implied that W0 and Y characterize the spin fluctuations and these are fixed parameters in the fitting procedure. 
I can see that W0 needs to be fixed because it was measured, but why does Y = W0/2, a simple even fraction, work fine for what would seem to be a sensitive parameter? 
Yes I add this explanation: "In any case, even making different choices for Y, it can be verified that as Y increases, the value of Ld decreases and there are no large Ld variations for reasonable Y choices. 
For example if Y=W0 the reduction of Ld is of the order of four percent respect to Y=W0/2. 
The trend as a function of Ld doping remains the same, just the coefficients in the function fit (Ld versus Ls) change a bit."

Finally, the fit of the theory to the data is excellent, with only one free parameter.
Does this imply that λd is determined from the fit?  
And yet it is related to λZ, which was measured. 
The values of Ld are determined in the way to reproduce the experimental critical temperature. The values obtained for Ld in fuction of λZ after are fitted by a analitical expression: the equ 10.

Something seems to be circular here, or in a way it shows the validity of Equation 10.  The conclusion is not clear to me. 
The equ 10 is just a simple fuction that reproduces the calculated values of ld. The conclusion is that a simple model as standard d wave Eliashberg equations can reproduce the experimental data and this means that this physical system is not so exotic. 

Round 2

Reviewer 3 Report

In this second revision I believe the manuscript is ready for publication. 

Author Response

In this latest version of the paper that I now submit, I think I have corrected and improved the English.